# Temporally and Spatially Resolved Reflected Overpressure Measurements in the Extreme Near Field

**DOI:** 10.3390/s23020964

**Published:** 2023-01-14

**Authors:** Andrew D. Barr, Sam E. Rigby, Sam D. Clarke, Dain Farrimond, Andy Tyas

**Affiliations:** 1Department of Civil and Structural Engineering, The University of Sheffield, Sir Frederick Mappin Building, Mappin Street, Sheffield S1 3JD, UK; 2Blastech Ltd., The Innovation Centre, 217 Portobello, Sheffield S1 4DP, UK

**Keywords:** blast waves, explosives, near field, pressure measurement, Hopkinson pressure bar, dispersion, Kingery–Bulmash, stress waves, shock, detonation products

## Abstract

The design of blast-resistant structures and protective systems requires a firm understanding of the loadings imparted to structures by blast waves. While empirical methods can reliably predict these loadings in the far field, there is currently a lack of understanding on the pressures experienced in the very near field, where physics-based numerical modelling and semi-empirical fast-running engineering model predictions can vary by an order of magnitude. In this paper, we present the design of an experimental facility capable of providing definitive spatially and temporally resolved reflected pressure data in the extreme near field (Z<0.5 m/kg${}^{1/3}$). The Mechanisms and Characterisation of Explosions (MaCE) facility is a specific near-field evolution of the existing Characterisation of Blast Loading (CoBL) facility, which uses an array of Hopkinson pressure bars embedded in a stiff target plate. Maraging steel pressure bars and specially designed strain gauges are used to increase the measurement capacity from 600 MPa to 1800 MPa, and 33 pressure bars in a radial grid are used to improve the spatial resolution from 25 mm to 12.5 mm over the 100 mm radius measurement area. In addition, the pressure bar diameter is reduced from 10 mm to 4 mm, which greatly reduces stress wave dispersion, increasing the effective bandwidth. This enables the observation of high-frequency features in the pressure measurements, which is vital for validating the near-field transient effects predicted by numerical modelling and developing effective blast mitigation methods.

## 1. Introduction

The design of structures to protect against high-explosive detonations has traditionally been of interest to military engineers, but the recent prevalence of terrorist attacks (5226 attacks in 2021 alone [1]) has also driven an increased need for engineers to consider the effects of these threats on civilian structures. Blast protection design requires a detailed knowledge of the loading imparted on a structure by a particular explosive threat, including the mechanisms involved in the rapid energy release that leads to fireball expansion and air shock development.

In the far field, where the explosion is sufficiently distant that only the propagating air shock interacts with the structure, reliable semi-empirical predictive methods exist for both the positive [2] and negative [3] phases of the blast wave for spherical and hemispherical charges. Well-controlled experimental studies using commercially available piezo-resistive or piezo-electric pressure sensors have repeatedly shown these methods to be accurate for scaled distances Z>2 m/kg${}^{1/3}$ [4,5,6,7,8].

In the near field, the fireball of detonation products is still expanding and driving the air shock and can itself interact with the structure. The Kingery–Bulmash predictions [2] used in ConWep [9] and other fast-running engineering models (FREMs) are not defined by direct measurements in the near field, and so the limited data at Z<0.4 m/kg${}^{1/3}$ contain significant scatter [10] and have been shown to diverge rapidly from physics-based numerical modelling predictions at small scaled distances (Figure 1a). The validation of numerical modelling approaches, and an understanding of the mechanisms of energy release during the early stages of detonation, therefore requires experimental data on the spatial and temporal distribution of pressure in the near field. The challenge for researchers is the incredibly high pressures and temperatures associated with near-field measurements, for which traditional pressure transducers are insufficiently robust. Alternative approaches such as measuring the momentum imparted to a free-flying plate or plug can be used to calculate the specific impulse [11] but cannot provide any information on the shape of the blast wave or the peak overpressure.

Hopkinson pressure bars (HPBs) instrumented with strain gauges are a robust method of providing this temporal resolution and have successfully been used by blast researchers in the near field [12,13,14,15], building on Hopkinson’s pioneering work over 100 years ago [16]. Of particular note, the Characterisation of Blast Loading (CoBL) facility [17] used a two-dimensional array of 10 mm diameter HPBs arranged in a stiff steel plate to record both the temporal and spatial variation of reflected pressure for buried [18] and free-air blasts [19]. The CoBL facility has been used at scaled distances 0.15≤Z≤0.75 m/kg${}^{1/3}$, spanning both the “extreme” near field (Z⪅0.5 m/kg${}^{1/3}$) and the “late” near field (0.5⪅Z⪅2 m/kg${}^{1/3}$). The extreme near field is defined as the range of stand-off distances where the loading from the nascent fireball is highly repeatable, while the late near field indicates the range where the development of Rayleigh–Taylor and Richtmyer–Meshkov instabilities in the fireball as it expands leads to a significant increase in variability in the loading observed [20].

The CoBL facility’s performance in the extreme near field is limited by several factors including HPB yield strength, strain gauge capacity and the effects of dispersion on HPB stress wave propagation [21], while the massive and permanent nature of its concrete construction also limits potential experimental configurations and the integration of advanced diagnostic techniques. This paper presents the design of a new Mechanisms and Characterisation of Explosions (MaCE) facility, a lab-scale evolution of CoBL with significantly increased pressure capacity, measurement bandwidth and spatial resolution. The MaCE facility incorporates 4 mm diameter high-strength maraging steel HPBs embedded in a stiff plate in a radial grid, with 12.5 mm resolution over a 100 mm radius measurement area. As well as enabling spatially and temporally resolved reflected overpressure measurements of free-air blasts at scaled distances below 0.1 m/kg${}^{1/3}$ for the validation of modelling approaches, the portability of the facility also enables the integration of advanced diagnostics such as fireball thermometry, chemical analysis and stereo DIC to interrogate the mechanisms of early fireball development and late near-field loading variability.

## 2. Experimental Design

### 2.1. Expected Loading Profile

The peak in-service loads experienced by a novel experimental facility for extreme near-field pressure measurement cannot, by definition, be confidently calculated; however, existing numerical methods can be used to provide an indicative design load for the scaled distances of interest. Figure 1b shows the peak reflected pressures predicted on a rigid surface from the detonation of a 100 g spherical charge of PE-4 at stand-off distances between 35 mm and 100 mm (as measured from charge centre). The Multi-Material Eulerian/Arbitrary Lagrangian–Eulerian solver in LS-DYNA was used to simulate the detonation, shock wave propagation and interaction with the rigid boundary, using a method previously described by the authors [22]. In each case, the charge was modelled in a 250×250 mm domain of 1 mm square axi-symmetric 2D-ALE elements, as informed by a preliminary mesh sensitivity study. Ideal gas behaviour was assumed for air, and PE-4 was modelled with *MAT_HIGH_EXPLOSIVE_BURN and *EOS_JWL, using the parameters tabulated in [22].

A 100 g sphere of PE-4 has a radius of approximately 25 mm, and so these scaled distances (0.08≤Z≤0.22 m/kg${}^{1/3}$) represent the extreme near field, approaching physical contact with the structure. The radial ordinates in Figure 1b indicate distances along the reflecting surface from the point which is normal to the charge centre. The peak reflected pressures directly in line with the charge (radial ordinate 0 mm) rise quickly as stand-off distance is reduced, approaching 3 GPa at Z=0.08 m/kg${}^{1/3}$, but in all cases, the predicted pressure reduces below 300 MPa at radial ordinates of 50 mm and above. This indicates that, while high-strength steels and special gauging methods will be required in the central region, economies can be made at larger radial ordinates. It should also be noted that the oscillations visible in some of these peak pressure predictions are likely to be non-physical, highlighting the difficulty of modelling these highly nonlinear processes.

### 2.2. Test Frame

Like the previous CoBL facility, the MaCE facility (Figure 2) eliminates the effects of structural compliance on measured loading magnitude and duration by adopting a rigid target plate, which is also sufficiently wide to avoid clearing effects over the expected loading duration [23,24]. A 50 mm thick, 850 mm diameter EN24T steel plate forms the main reflecting surface, with a replaceable 25 mm thick, 270 mm diameter 300M steel plate in the instrumented central area, where pressures are highest. Provision for buried blast assessment is not required as it was in CoBL, and so the supporting structure can be significantly simplified into a portable lab-scale frame formed of aluminium extrusions, with the target plate fixed on top. The rigidity of the main target plate is ensured by the short 400 mm span between its supports, and loads are directly transmitted through the stiff frame to the ground. The frame is fitted with shock-absorbing rubber feet, which also have retractable heavy-duty castor wheels for manoeuvrability. The pressure bars and electronics are housed inside the frame, and are protected from the effects of the air shock by removable aluminium side panels. Buckling of the pressure bars is prevented by a series of lateral supports fitted with bushings to enable free axial movement and shock absorbers at the base of each bar. The main target plate also features fastening points for a removable blast chamber, which are used to investigate the effects of afterburn through control of the atmospheric gases around the explosive charge [25], and enable thermal [26] and chemical [27] analysis of the detonation products.

### 2.3. Hopkinson Pressure Bar Design

The design of a Hopkinson pressure bar with surface-mounted strain gauges requires careful consideration for near-field blast measurement, as the bar material, bar geometry and strain gauge specification all have a significant impact on the load capacity, measurable load duration and measurement bandwidth of the resulting instrument.

The initial modelling study above indicated that the peak reflected pressures could exceed several gigapascals in the extreme near field. As the analysis of HPB signals relies on the propagation of elastic stress waves in the bar, the first limit imposed is the yield strength of the bar material. The bars in the central 50 mm radius of the target plate are specified from grade 350 maraging steel, which in its hardened condition has a 0.2% proof stress of at least 2200 MPa. Bars at more than 50 mm radius from the centre of the target plate are expected to experience much lower peak pressures and so are constructed from F51 stainless steel, which has a 0.2% proof stress exceeding 450 MPa.

Pressure signals in the extreme near field are also expected to contain significant high-frequency content due to short rise times and rapid pressure transients. The velocity of stress waves in a pressure bar is a function of frequency, with higher frequency components propagating more slowly than lower frequency components. This leads to a phenomenon called dispersion, where the relative movement of these components results in a change in the shape of the signal between the face of the bar and the strain gauge location [21]. Techniques have been developed to account for the effects of dispersion in pressure bar signals [28,29], but limits on bandwidth are still imposed by the variation of stress over the bar cross section, and the occurrence of “nodal cylinders” on the bar surface at some frequencies, where zero axial strain is recorded on the bar surface despite a non-zero internal strain. One of the simplest ways to minimise dispersive and cross-sectional effects is by increasing the ratio of the signal wavelength to pressure bar radius, that is, using a smaller diameter pressure bar. The MaCE bar array is formed of 4 mm diameter pressure bars in comparison to CoBL’s 10 mm bars, which delays the appearance of the first nodal cylinder in the bar from 300 kHz to over 700 kHz, significantly increasing the effective measurement bandwidth.

A further limit on the load capacity of the bars is the strain measurement limit of the strain gauges. Semiconductor gauges (Kyowa KSPB-2-120-E4) were previously used in the CoBL apparatus, as their high gauge factors and short active length provided a good signal–noise ratio and frequency response when compared to traditional foil gauges [17]. However, the 3000 μm
$m^{-1}$ strain capacity of these gauges limits the measurable pressure at around 600 MPa, and the surface curvature of a 4 mm pressure bar complicates the application of the gauges, as the semiconductor element is mounted on a larger polyimide membrane. The MaCE apparatus instead uses smaller bare semiconductor gauges without a backing membrane, which can be precisely applied to the bar surface using epoxy. Bars at more than 50 mm radius from the centre of the target plate are fitted with Micron SS-027-013-500P gauges, which have a 3000 μm
$m^{-1}$ strain capacity (approximately 600 MPa). Bars in the centre of the target plate are fitted with “crash” gauges supplied by Haptica S.r.l.: the special geometry of these Micron SS-040-010-1100P gauges increases their strain capacity to 9000 μm
$m^{-1}$ (approximately 1800 MPa), enabling the investigation of blast pressures in the extreme near field. The sub-millimetre length of the bare strain gauges makes their application challenging, and so these were installed on the bars by the manufacturer. An optical microscope was used to aid precise placement and orientation: Haptica reports typical alignment tolerances of less than 2 degrees, meaning that the associated measurement errors should be negligible. A comparison of the properties of the strain gauges is shown in Table 1.

Two gauges are installed axially on the surface of each bar, diametrically opposed in order to eliminate strains due to bending. The gauges are located 40 mm from the top face of the 1 m long bars to minimise dispersive effects, providing a maximum recording time of approximately 0.38 ms before reflections from the distal end of the bar begin to interfere with the incident signal. A Wheatstone bridge circuit for each bar is populated with these two active gauges, and two additional temperature compensation gauges, which are bonded to a small sample of bar material inside the electronics enclosure. This ensures that changes in gauge resistance due to changes in ambient temperature are automatically accounted for in balancing the bridge circuit, while the temperature dependence of the gauge factors (see TCGF, Table 1) can be corrected for algorithmically. The high gauge factors of the semiconductor gauges mean that additional amplification in the circuit is unnecessary, and so the output voltages of the bridge circuits are recorded directly. A bridge excitation of 5 V is provided by a bench power supply, and pressure bar signals are recorded using TiePie Handyscope HS6 differential oscilloscopes in 14-bit resolution, at a sampling rate of 5 MHz. Nine four-channel oscilloscopes are required to accommodate the 33 signal channels, but the native multi-instrument synchronisation ensures a common timebase accuracy of 0 ppm.

### 2.4. Hopkinson Pressure Bar Array

A key feature of the CoBL apparatus was the ability to measure the reflected pressure history at multiple points on the target plate, meaning that the spatial variation of pressure could be determined. This took the form of 17 pressure bars arranged in a cross-shaped array: one central bar and two perpendicular axes covering the 100 mm radius instrumented area at a bar spacing of 25 mm. Investigation of the extreme near field requires an even higher spatial resolution, and so the MaCE apparatus decreases the pressure bar spacing to 12.5 mm, as shown in Figure 3a. To enable comparisons with existing CoBL data, four radial bars at 25 mm spacing are retained at 90 degree increments around the central bar. Additional bars are introduced along the diagonals, with four radial bars at 25 mm spacing starting 12.5 mm from the central bar, for a total of 33 pressure bars. This arrangement strikes a balance between increased spatial resolution, angular coverage of the target plate and the number of measurement points at the same radial ordinate, enabling pressure distribution plots of the target plate to be produced using previously-developed interpolation methods [17]. The polar coordinate system used to reference the bar locations is shown in Figure 3b.

## 3. Experimental Measurements

To demonstrate the improved performance of the new MaCE pressure bars, this section provides a comparison between CoBL and MaCE facility pressure measurements in the extreme near field. These tests were performed at Z=0.18 m/kg${}^{1/3}$, near the limit of the existing CoBL pressure bars, using the plastic explosive PE10 (86% PETN, 14% plasticiser; TNT${}_{\mathrm{eq}}\;=\;1.22$ [30]). Separate tests were performed in CoBL and MaCE using the same charge arrangement: a 113 g sphere of PE10 was positioned at a stand-off distance of 95 mm (charge centre to target plate surface) by suspending it between two vertical supports in a strip of lightweight glass fibre netting. In each case, the pressure signals were processed to account for the effects of dispersion between the loaded end face of the pressure bars and the strain gauge location on the bar surface using dispersion.m [31]. This frequency-domain algorithm corrects both the phase angle and amplitude of frequency components (within the pressure bar’s correctable bandwidth) so that the processed signals more closely represent the stresses experienced on the reflecting surface of the target plate [28].

Figure 4 shows a frequency-domain representation of the signals recorded on the central pressure bars in MaCE and CoBL, along with the key relationships required for dispersion correction. Phase angle corrections are applied to each frequency component based on the ratio of the component’s phase velocity, $c_{p}$, to the bar material’s one-dimensional wavespeed, $c_{0}$ (Figure 4a). Amplitude corrections (Figure 4b) are applied to account for the differences between the mean cross-sectional strain and the strain measured on the bar surface (factor $M_{1}$) and between the mean cross-sectional strain and mean cross-sectional stress (factor $M_{2}$, normalised by Young’s modulus, *E*). At some frequencies, a nodal cylinder exists on the surface of the bar, and no axial strain is measured despite a non-zero strain inside the bar: the first such nodal cylinder occurs at $fa/c_{0}=0.30$, where *a* is the bar radius. Since no signal information can be recorded at this frequency, the first nodal cylinder presents a practical limit on the bandwidth of dispersion correction, as the original loading cannot be reconstructed. Figure 4c shows the power spectral density of the signals recorded in the MaCE and CoBL facilities, along with the frequencies at which nodal cylinders occur on the bar surface. The first nodal cylinder in the CoBL pressure bar was calculated to occur at approximately 310 kHz and can be identified by the distinct drop in the power of the signal at this frequency. The smaller radius of the MaCE pressure bars means that frequencies correspond to lower normalised frequencies in Figure 4a,b, and so the nodal cylinder does not occur until approximately 735 kHz, significantly reducing the effects of dispersion and expanding the bandwidth over which dispersion correction can be applied.

Figure 5a,b show the overpressures measured by the central pressure bar following detonation in the CoBL and MaCE facilities, respectively. The measurements from the 10 mm diameter pressure bars (Figure 5a) in CoBL are dominated by large oscillations at this scaled distance, which indicates significant stress wave dispersion as a result of high-frequency components in the signal and the bars’ larger diameter. The lower bandwidth of these bars also limits the extent to which dispersion correction can resolve the original loading at the bar face: rise time is slightly decreased and peak pressure slightly increased, but the remaining oscillations will obscure any physical high-frequency features in the signal. In contrast, while there is also some evidence of dispersion in the 4 mm diameter MaCE bars (Figure 5b), most of the energy of the signal propagates at frequencies where dispersion correction can be applied successfully, and spurious oscillations in the pressure measurement can largely be removed.

Comparisons of the dispersion-corrected reflected pressure in MaCE and CoBL are shown in Figure 5c, where significant differences can be observed between the measurements. The increased bandwidth of the MaCE pressure bars enables high-frequency features to be recorded, such as the rapid rise to the initial high-pressure peak, and so MaCE records a peak reflected pressure of 372 MPa. This is 70% higher than the 220 MPa peak reflected pressure recorded by CoBL. Similarly, the majority of the MaCE loading occurs in the first 25 μs, while the dispersion of the CoBL signal implies a positive phase duration of over 50 μs even after correction is attempted. Figure 5d shows a comparison of specific impulse, which indicates that the peak specific impulses are very similar in both cases at approximately 3.6 MPa
ms. This similarly is to be expected, as both the 4 mm and 10 mm diameter bars should experience the same total impulse; however, the dispersion of the stress waves in the larger-diameter 10 mm bar results in an elongation of the stress pulse, meaning that specific impulse at the measurement location accumulates more slowly than in the 4 mm bar.

Also shown in Figure 5c,d are the semi-empirical Kingery–Bulmash predictions [2], calculated using blast.m [32]. The predicted peak specific impulse matches the experimental data quite well (3.7 MPa
ms), but as the Kingery–Bulmash parameters are not defined by direct experimental measurements at these scaled distances, reflected pressure is significantly underpredicted (177 MPa) and positive phase duration is significantly overpredicted (90 μs). Just like the heavily dispersed CoBL signals, this inaccuracy in the pressure–time history results in the predicted specific impulse rising more slowly than in the experimental data. Some engineering problems in blast can be adequately addressed with a good prediction of peak specific impulse, and if the Kingery–Bulmash impulse retains this accuracy further into the extreme near field, FREM predictions (Figure 1a) will remain useful for many structural analyses. However, the development and validation of models for loading in the extreme near field also requires an accurate understanding of the temporal development of overpressure.

The MaCE facility’s ability to record high-frequency features is crucial for investigating the mechanisms involved in the rapid energy release at these early stages of detonation, as an understanding of fireball expansion and air shock development is a prerequisite for developing effective blast mitigation methods. An example of such a high-frequency feature is the large transient peak pressure in Figure 5b, which resembles observations in an earlier experimental and numerical study by Edwards et al. [12]. This work attributed high initial pressures observed in the extreme near field to repeated reflections of the air shock between the target plate and the boundary with the dense detonation products. The superimposed second shock appeared as a “knee” in the initial rise, which also appears to be visible in the current measurements in Figure 5b. The study noted limitations due to wave dispersion and bar yield strength, and so further experimental programmes with the MaCE facility and dispersion–correction methods will enable the full spatial distribution of reflected pressure to be analysed in the extreme near field. These experiments will confirm the mechanisms driving early post-detonation behaviour, which will be especially important for more complex scenarios such as the analysis of non-ideal explosives or the effects of initial atmospheric conditions on afterburn.

## 4. Conclusions

This paper presents the design of an experimental facility for the measurement of reflected overpressures from blast loading in the extreme near-field. Following an initial numerical modelling study, high-strength maraging steel bars and specially-designed semiconductor strain gauges were used to develop Hopkinson pressure bars capable of reflected pressure measurements of at least 1800 MPa, which the modelling predicts to correspond to a scaled distance of less than 0.1 m/kg${}^{1/3}$. In order to minimise the effects of stress wave dispersion a small 4 mm pressure bar diameter was selected, and the strain gauges were placed on the bar surface 40 mm from the loaded face. These changes reduced dispersive effects and increased the bandwidth over which dispersion correction can be applied during signal processing, from approximately 300 kHz in the previous CoBL facility to over 700 kHz in the new design.

The pressure bars were housed in a portable lab-scale test frame, with a 50 mm thick, 850 mm diameter steel target plate to provide a rigid surface for reflected pressure measurements. The construction of this frame also allows for the addition of a removable blast chamber, which enables thermal and chemical analysis of the detonation products and control over the atmospheric gases. A total of 33 Hopkinson pressure bars were arranged in a radial grid to provide high spatial resolution (12.5 mm) of reflected pressures over a 100 mm radius, enabling interpolated plots of both the spatial and temporal distribution of stress.

Identical free-air blast experiments were used in the MaCE and CoBL facilities to demonstrate the improved performance of the MaCE pressure bars in the extreme near field (Z=0.18 m/kg${}^{1/3}$). The 4 mm diameter MaCE bars exhibited greatly reduced wave dispersion when compared to the 10 mm CoBL bars, which enabled the identification of a transient high-pressure peak and the positive phase duration. Both the CoBL experiment and Kingery–Bulmash predictions provided good values for peak specific impulse but significantly underpredicted peak pressure and overpredicted the loading duration, meaning that the impulse accumulated more slowly.

The improved ability of the MaCE pressure bars to record these high-frequency features of the blast loading accurately is of particular importance for developing accurate models of blast loading and mechanisms of energy release in the extreme near field. With the addition of the chemical and thermal diagnostics described above, this facility will be used to investigate early fireball expansion and air shock development, informing accurate models of early post-detonation behaviour.

## Figures and Tables

**Figure 1 sensors-23-00964-f001:**
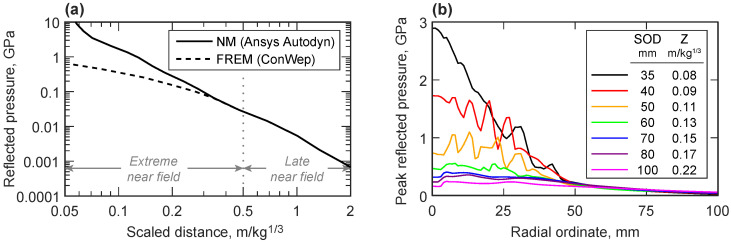
Predictions of near-field reflected overpressures. (**a**) The large discrepancy in the predictions made by numerical models such as Ansys Autodyn [10] and fast-running engineering models such as ConWep [9] in the extreme near field. (**b**) Numerical predictions for the peak reflected overpressure with radial ordinate on a rigid reflecting plate, resulting from a 100 g spherical charge of PE-4 at various stand-off distances.

**Figure 2 sensors-23-00964-f002:**
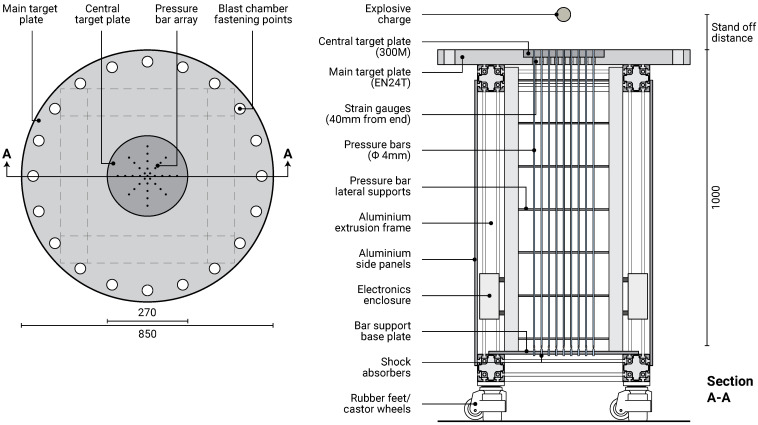
MaCE near-field facility in plan view and section. Dimensions are in mm.

**Figure 3 sensors-23-00964-f003:**
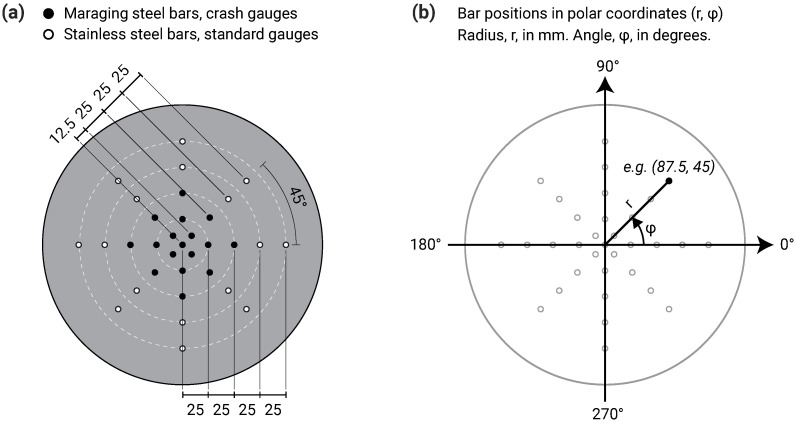
Central target plate pressure bar array, covering a circle with radius 100 mm. (**a**) Pressure bars within the inner 50 mm radius use high strength maraging 350 steel and specially manufactured “crash” gauges. Pressure bars outside the inner 50 mm radius use F51 stainless steel and standard strain gauges. (**b**) Polar coordinate system used to reference the bar positions, with angles measured in degrees, anticlockwise from right.

**Figure 4 sensors-23-00964-f004:**
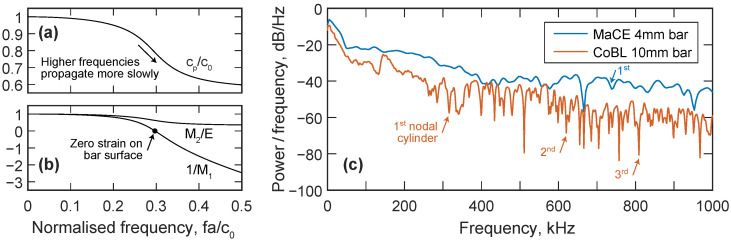
(**a**) Relationship between phase velocity, $c_{p}$, and one-dimensional wavespeed, $c_{0}$, with varying normalised frequency for the first mode of propagation (ν=0.29). (**b**) Values of $1/M_{1}$ and $M_{2}/E$ with varying normalised frequency for the first mode of propagation. The labelled point indicates the frequency where a nodal cylinder exists on the surface of the bar: zero strain is recorded on the bar surface despite a non-zero internal strain. (**c**) Power spectral density of MaCE and CoBL measurements indicating the frequencies where nodal cylinders occur.

**Figure 5 sensors-23-00964-f005:**
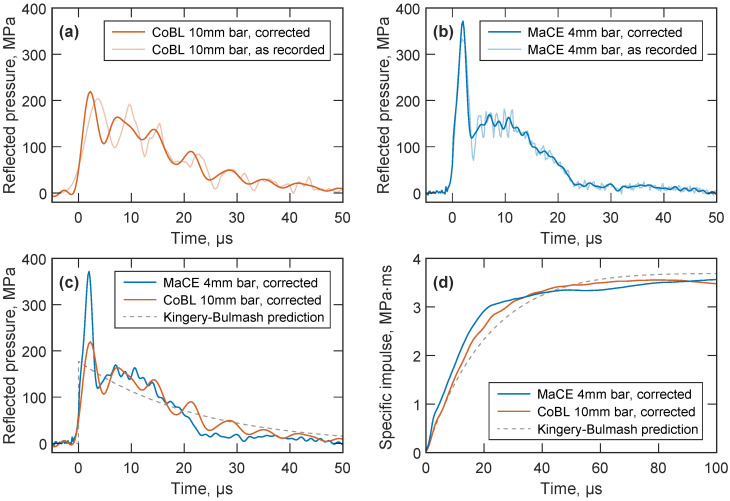
Reflected pressure measurements for a spherical PE10 charge at Z=0.18 m/kg${}^{1/3}$, using (**a**) 10 mm diameter pressure bars in CoBL and (**b**) 4 mm diameter pressure bars in MaCE. (**c**) Reflected pressure and (**d**) specific impulse comparisons of dispersion-corrected MaCE and CoBL data, and the semi-empirical Kingery–Bulmash predictions.

**Table 1 sensors-23-00964-t001:** Nominal properties of the semiconductor strain gauges used on the CoBL (Kyowa) and MaCE (Micron) pressure bars.

Strain Gauge	Max Strain, μm${}^{-1}$ (max MPa)	Active Length, mm	Resistance at 25 °C, Ω	Nominal Gauge Factor	TCGF ^1^, %°C	TCR ^2^, %/°C
Kyowa KSPB-2-120-E4	3000 (600)	2.00	120	125	−0.20	0.17
Micron SS-027-013-500P	3000 (600)	0.33	540	155	−0.32	0.43
Micron SS-040-010-1100P	9000 (1800)	0.25	1100	200	−0.41	0.72

^1^ TCGF = Temperature coefficient of gauge factor, ^2^ TCR = Temperature coefficient of resistance.

## Data Availability

MATLAB software used in this study is openly available in public repositories: dispersion.m is available on FigShare at https://doi.org/10.15131/shef.data.3996876.v1 (accessed on 12 January 2023); blast.m is available on White Rose Research Online at https://eprints.whiterose.ac.uk/77858/ (accessed on 12 January 2023).

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
