# Peer review of "Temporally and Spatially Resolved Reflected Overpressure Measurements in the Extreme Near Field"

_sensors, 2023, doi:10.3390/s23020964_

Round 1

Reviewer 1 Report

Article Summary

The manuscript presents a significant update on an experimental rig using Hopkinson Pressure Bars to measure shock waves due to near field explosive loading. The updates include improved spatial resolution and higher frequency bandwidth due to changes in HPB parameters. A limited set of results, comparing the pressure histories for the updated HPB system to the prior version for nominally identical explosive loading, is presented and some of the key features and differences discussed.

Rating of Manuscript

I believe the manuscript merits publication, with some minor suggestions for improvements.

While the concept applied for the measurements is not novel, the authors have leveraged some technical advances in strain gauges and design decisions around the HPB array to make a significant improvement over the previous iteration of this experimental rig, specifically in improving frequency bandwidth and spatial resolution of pressure. Accurate quantification of blast wave pressures in the extreme near field remains an ongoing research area for engineers studying explosive loading and structural response thereto. The advances presented in this manuscript are relevant to the journal scope (Special Edition: Metrology of  Shock Waves).   

While the limited data set presented in the manuscript isn’t hugely significant on its own, it’s clear that the rig will be used in broader experimental study that will yield novel and significant data. In addition, the results presented could on its own be used for preliminary verification of numerical simulations of explosive loading. The description of the design revisions and preliminary results will be of interest to both researchers in the field of explosive loading, and design engineers who need to better quantify the loads imposed on structures.

The manuscript is clearly written with appropriate tone and grammatically correct English.

General Comments

Figure 1 is used to illustrate that the peak overpressures presented by FREMs and numerical models are divergent for the extreme near field. The x-axis (Scaled distance) is presented in log scale, while the y-axis (Reflected pressure) is a linear scale. This does show the extreme divergence at very small scaled distance. I’d encourage the authors to re-plot this with the y-axis as a log scale, to see if this allows the divergence to be more clearly visible over a greater range of scaled distance.

Instrumentation Description

Sec 2.3 provides a well detailed description of the strain gauges utilised, and the limitations this imposes on the measurements. The strain gauge arrangements and oscilloscopes used to measure these are sensible. However, there is some information missing that would be very helpful:

-Given these are semiconductor strain gauges with large gauge factors, I am assuming that there is no additional amplification between the Wheatstone bridge and the oscilloscope? A clarification of this would be greatly appreciated!

-The range of bridge excitation voltages isn’t mentioned, but would be of interest to researchers working on similar experiments.

Calibration of Strain Gauges & Pressure Bars

There isn’t any description of a calibration procedure. Given the very small strain gauges and the bar radius, I suspect it’s very challenging to bond the strain gauges to the bars with their respective axes aligned accurately. Without requiring a detailed description of gauge bonding or calibration procedures, it would be good to know if the COBL and MACE HPBs are calibrated similarly and whether any gauge misalignment is corrected for during calibration. 

Presentation of Results

A single set of pressure data is presented for the older COBL and newer MACE rigs. It’s not clear if this data was for the same explosive event, or if both rigs were subjected to separate explosions with identical charge mass and standoff distance. While the agreement for lower pressures is good, it would be nice to know if we can ignore any variation between detonations.

The pressure data presented is for the central HPB. Was data captured for other bars at off-centre locations? Even if the location of the bars is different for the COBL and MACE rigs, it would be very interesting to see spatial variation of pressure.

The pressure data presented is all time domain data. One of the key features of the update is improvements in the frequency domain. If the same pressure data sets were presented in the frequency domain (perhaps alongside a phase velocity-frequency plot for the dispersion correction map?), it would allow stronger arguments about the gains made, as well as any decisions about filtering and frequency cut-offs.

Author Response

The authors thank the reviewer for their prompt review and insightful comments on our manuscript. Please see the attachment for our responses.

Reviewer 2 Report

It is not clear whether the development is theoretical and validated with numerical models (vaguely explained and basically referring to previous publications) or built-in reality and tested in real trials. If the latter term is the correct one, I do not understand why no image of the actual device, or of the tests performed, is published.

More details of the actual tests (if they have been performed) and of the numerical model (not only mentioning references of the authors) should be provided in the text.

why the radial spacings in figure 3 are every 25 mm? would it change the efficiency of the system if they were spaced at a different interval?

It would be convenient to explain briefly how the dispersion.m program works, in terms of relations or geometries used in the corrections.

In the phrase “Kingery–Bulmash parameters are not defined by direct experimental measurements at these scaled distances” please add the validity (field test) range of the K&B relations.

The link in reference 32 does not work.

Author Response

The authors thank the reviewer for their prompt review. Please see the attachment for our responses.

Round 2

Reviewer 2 Report

It can be accepted